https://doi.org/10.1038/s41467-019-08781-2　　**OPEN**

# Promiscuous enzymatic activity-aided multiple-pathway network design for metabolic flux rearrangement in hydroxytyrosol biosynthesis

Wei Chen[1], Jun Yao[1,2], Jie Meng[1,2], Wenjing Han[3], Yong Tao[1], Yihua Chen[1,2], Yixin Guo[4], Guizhi Shi[2], Yang He[5], Jian-Ming Jin[3] & Shuang-Yan Tang[1]

Genetic diversity is a result of evolution, enabling multiple ways for one particular physiological activity. Here, we introduce this strategy into bioengineering. We design two hydroxytyrosol biosynthetic pathways using tyrosine as substrate. We show that the synthetic capacity is significantly improved when two pathways work simultaneously comparing to each individual pathway. Next, we engineer flavin-dependent monooxygenase HpaBC for tyrosol hydroxylase, tyramine hydroxylase, and promiscuous hydroxylase active on both tyrosol and tyramine using directed divergent evolution strategy. Then, the mutant HpaBCs are employed to catalyze two missing steps in the hydroxytyrosol biosynthetic pathways designed above. Our results demonstrate that the promiscuous tyrosol/tyramine hydroxylase can minimize the cell metabolic burden induced by protein overexpression and allow the biosynthetic carbon flow to be divided between two pathways. Thus, the efficiency of the hydroxytyrosol biosynthesis is significantly improved by rearranging the metabolic flux among multiple pathways.

[1] CAS Key Laboratory of Microbial Physiological and Metabolic Engineering, State Key Laboratory of Microbial Resources, Institute of Microbiology, Chinese Academy of Sciences, Beijing, China. [2] University of Chinese Academy of Sciences, Beijing, China. [3] Beijing Key Laboratory of Plant Resources Research and Development, Beijing Technology and Business University, Beijing, China. [4] Center for Drug Discovery & Technology Development of Yunnan Traditional Medicine, Yunnan Provincial Academy of Science and Technology, Kunming, China. [5] Life Science Institute, Zhejiang University, Hangzhou, China. These authors contributed equally: Wei Chen, Jun Yao. Correspondence and requests for materials should be addressed to Y.H. (email: younigh@zju.edu.cn) or to J.-M.J. (email: jinjianming@btbu.edu.cn) or to S.-Y.T. (email: tangsy@im.ac.cn)

Metabolic engineering is a powerful tool for over-production of a huge number of natural products whose natural resources are limited[1]. Improvement of the biosynthetic pathway efficiency is one of the main goals of metabolic engineering[2]. In common, it was achieved by over-expression or engineering of rate-limiting enzymes[3,4], balancing of the relative expression levels of the pathway enzymes[5,6], regeneration of cofactors[7], and so on.

In nature, some steps of natural product biosynthetic pathways occur in alternative orders, for example, the methylation steps in gentamicin biosynthesis[8,9], the glycosylation steps in glycosyl-phosphatidylinositol biosynthesis[10], the methylation and hydroxylation steps in erythromycin A biosynthesis[11]. Besides the generation of more compounds to adapt to environmental variations, another possible advantage of these multiple-pathway strategies is increasing the efficiency of product formation. In laboratory engineering, if we build a network that uses a same set of enzymes to convert a substrate to a product via multiple pathways as the cases in nature (Fig. 1), will it improve the productivity as anticipated? Herein we used the engineering of hydroxytyrosol biosynthesis as an example to explore this idea.

Hydroxytyrosol is a powerful antioxidant scavenger of free radicals that confers cell protection[12,13]. The hydroxytyrosol biosynthetic pathway, which uses tyrosine as a substrate, was first reported by Satoh[14]. In this pathway, the tyrosine hydroxylation step catalyzed by mouse tyrosine hydroxylase using cofactor tetrahydromonapterin (MH4) severely limits biosynthetic efficiency. Another pathway in which hydroxytyrosol was produced from tyrosine through the intermediate tyrosol was reported recently and the yield was close to 50%[15].

In pathway engineering, artificial pathways are sometimes designed to bypass rate-limiting steps or allow a shortcut for synthesis. We, therefore, attempt to design hydroxytyrosol biosynthetic pathways for more efficient biosynthesis in *E. coli*. Two pathways are designed. In each pathway, there is a catalytic step lack of highly efficient enzymes. To fulfill these missing steps, two different mutant enzymes are evolved from a monooxygenase, individually. Both pathways exhibit elevated hydroxytyrosol yield compared with the reported pathways. Additionally, we test the hydroxytyrosol biosynthetic efficiency when both pathways co-work. To minimize the metabolic burden induced by protein overexpression, a monooxygenase mutant with promiscuous functions, which catalyzes both missing steps in the two pathways, is developed. And we find that the hydroxytyrosol yield is further improved significantly.

Our results provide a proof-of-concept design of a multiple-pathway network making use of an enzyme with designed promiscuity, which mimics the natural strategy and leads to significantly improved natural product biosynthesis efficiency.

## Results

**Design of biosynthetic pathways for hydroxytyrosol.** Due to the low biosynthetic efficiency of hydroxytyrosol in *E. coli* as previously reported, two pathways were designed. As shown in Fig. 2, in pathway 1, tyrosine decarboxylase (TDC) catalyzed the conversion of tyrosine to tyramine, which was then oxidized to 4-hydroxyphenylacetaldehyde (4HPAA) by tyramine oxidase (TYO). 4HPAA was reduced to tyrosol by alcohol dehydrogenase (ADH). Tyrosol was then hydroxylated into hydroxytyrosol by a tyrosol hydroxylase. Highly specific tyrosol hydroxylases have not been found. It has been reported that the monooxygenase HpaBC could catalyze the tyrosol hydroxylation[16], however, the activity was not sufficient, as tyrosol is not the optimal substrate of wild-type HpaBC and the tyrosol hydroxylation activity was relatively low.

Interestingly, because TDC, TYO, and ADH exhibit promiscuous substrate specificity, it is possible that hydroxytyrosol can also be synthesized through pathway 2. However, the step of dopamine synthesis from tyramine is missing. Although tyramine hydroxylase exists in mammals and plants, these enzymes are cytochrome P450s and are not expressed well or do not function well in prokaryotes[17–19].

To complete each of these two pathways, highly active tyrosol hydroxylase or tyramine hydroxylase was obtained through protein engineering described as follows.

**Development of tyrosol or tyramine hydroxylase activity.** The two-component flavin-dependent monooxygenase, HpaBC, has been found in *Escherichia coli* W, C and B strains[20], as well as in *Pseudomonas aeruginosa*[16] and *Klebsiella pneumoniae*[21]. As the first enzyme in the 4-hydroxyphenylacetate (4-HPA) degradation pathway, HpaBC catalyzes the conversion of 4-HPA to 3,4-dihydroxyphenylacetate (HPC)[22]. As shown in Supplementary Fig. 1, HpaBC exhibits a broad substrate spectrum, including chloro- and methyl-aromatic compounds, which suggests its potential for degrading environmental pollutants[20,23]. Wild-type HpaBC from *E. coli* B showed little activity on tyramine, as well as a minor activity on tyrosol that was lower than 10% of the activity on 4-HPA. It also showed little activity for converting tyrosine to dopa, which makes it a good candidate for hydroxytyrosol biosynthesis from tyrosine. In this study, protein engineering was used to expand the substrate specificity of this HpaBC from 4-HPA to tyrosol or tyramine, as shown in Fig. 3a.

We first aligned the sequences of HpaBs from various bacteria, and found they were conserved, especially for catalytic residues R113, Y117, and H155 (Supplementary Fig. 2A). Based on the alignment, we generated a molecular model to examine the substrate binding and catalytic mechanism of HpaB and we found that the overall structure of HpaB is quite identical as those from *Streptomyces globisporus*[24] and *Thermus thermophiles*[25] (Supplementary Fig. 2B). The Ramachandran plot for HpaB model is shown in Supplementary Fig. 2C and the substrate binding pocket comparison is shown in Supplementary Fig. 2D. 4-HPA is bound to a hydrophobic pocket formed by Y117, H155, Y301, V158, I157, and bound FAD. The distance from 4-HPA to FAD is 4.5 Å that is comparable to the oxygenase in *Thermus thermophiles* (Fig. 4a). Specifically, the hydroxyl group of 4-HPA forms hydrogen bonds with side chains of R113, Y117, and H155. Those three residues are thought to anchor the head of 4-HPA and mainly involved in catalytic intermediate stabilization. The

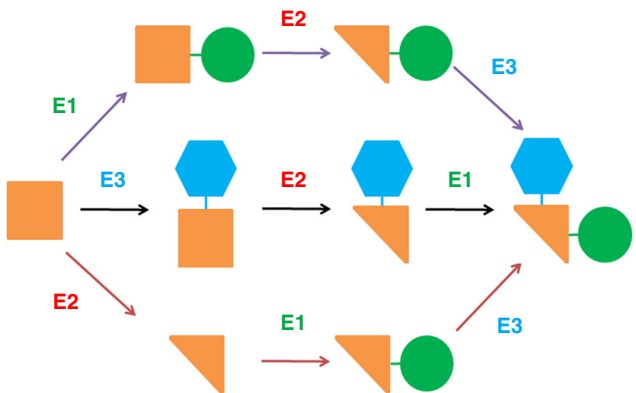

**Fig. 1** A network consisting of multiple pathways converting a substrate to a product using the same set of enzymes. The catalytic steps occur in alternative orders. The square, triangle, circle and hexagon represent the functional groups of a compound

**Pathway 1**

**Pathway 2**

**Fig. 2** Design of hydroxytyrosol biosynthetic pathways 1 and 2. The dashed arrows indicate the steps lacking in highly efficient enzymes. TDC, TYO, and ADH are used to design both pathways. The dashed arrows indicate the steps lacking in highly active tyrosol hydroxylase or tyramine hydroxylase

carboxyl group of 4-HPA is hydrogen-bonded to the main chain nitrogen atom of S210 and A211, and these two residues are likely to stabilize the tail of 4-HPA and determine the substrate selectivity. We hypothesized that the flexible loop containing S210 and A211 will select the substrate of HpaB through hydrogen bond formation with the tail of substrate. Accordingly, we choose S210, A211, and Q212 for simultaneous saturation mutagenesis, to screen the HpaB mutants which can recognize tyrosol or tyramine without any disturbance in catalytic activity.

To facilitate library screening, we designed a method for high-throughput screening of the hydroxylase activity of HpaBC. As shown in Fig. 3b, when reacted with sodium periodate, an oxidant chemically converting *o*-diphenol to *o*-quinone[26,27], only the hydroxytyrosol and dopamine reaction mixtures turned yellow. No color change was observed with tyrosine, tyrosol, and tyramine. This colorimetric difference could be quantified by measuring the absorbance ($OD_{400}$). As depicted in Fig. 3c, the $OD_{400}$ showed a positive linear correlation with the concentration of hydroxytyrosol or dopamine, the expected hydroxylated product of HpaBC mutants, whereas the substrate, tyrosol or tyramine, respectively, did not show any absorbance at 400 nm. Therefore, for high-throughput screening of HpaBC activity, the sodium periodate assay in a 96-well plate was developed (Fig. 3d).

To obtain mutants specifically reacting with tyrosol to produce hydroxytyrosol (tyrosol hydroxylase), the HpaBC saturation mutagenesis library was first screened in the presence of tyrosol for optimal activity. The selected mutants were subsequently screened in the presence of tyramine for negative activity. Seven mutants were obtained from a total of $10^5$ mutants screened, which showed significantly higher activities on tyrosol compared with the wild-type enzyme, but did not accept tyramine as a substrate. Mutant A10 showed the highest activity, exhibiting ~19-fold higher activity on tyrosol than the wild-type enzyme, as shown in Fig. 3e. Similarly, to obtain mutants specifically catalyzing tyramine to form dopamine (tyramine hydroxylase), the highest activities towards tyramine and subsequently the negative activities towards tyrosol were screened, using the same site-saturation mutagenesis library. Whereas wild-type HpaBC

exhibited extremely low activity on tyramine, seven mutants showing significant activities on tyramine were obtained, among which mutant D11 exhibited a ~386-fold higher activity on tyramine compared with the wild-type enzyme, as shown in Fig. 3f.

**Comparison of biosynthetic efficiency of pathway 1 and 2.** Pathway 1 or 2 was constructed via a plasmid expressing an operon consisting of genes encoding TDC, TYO, and HpaBC, as well as the ADH activity from the host cell. To optimize the hydroxytyrosol biosynthetic efficiency, we optimized the copy number of the plasmid and the order of the genes in the operon (Fig. 5). The plasmid P3, with an RSF3010 replication origin, yielded the highest amount of hydroxytyrosol (Fig. 5a). Furthermore, we explored the effect of the order of three genes in the operon, i.e., *tdc*, *tyo*, and the mutated *hpaBC* gene encoding the A10 mutant, on hydroxytyrosol production. The operon in plasmid P6 produced the highest amount of hydroxytyrosol (Fig. 5a).

In stain BHYT, which harbors plasmid P6 (Fig. 5a) expressing different HpaBC mutants, we determined the time course of the production of hydroxytyrosol and pathway intermediates as well as the consumption of tyrosine. The strain harboring plasmid P6 expressing wild-type HpaBC was used as a control. For the control strain (Fig. 5b), tyrosol was the primary final product, and tyramine was accumulated in large amounts because the steps needed to convert tyrosol to hydroxytyrosol and tyramine to dopamine were missing. TYO activity on tyramine was also not high enough in the control strain, leading to tyramine accumulation. In pathway 1, mutant A10 catalyzed tyrosol to produce hydroxytyrosol efficiently. Thus, there was no tyrosol accumulation, and the hydroxytyrosol concentration continued to increase through the entire culturing process. However, compared to the control strain, a slightly lower amount of tyramine accumulated, again indicating inadequate TYO activity in pathway 1 (Fig. 5c). In pathway 2, tyramine accumulation decreased significantly, implying that mutant D11 could efficiently convert tyramine to dopamine (Fig. 5d). Moreover, no dopamine

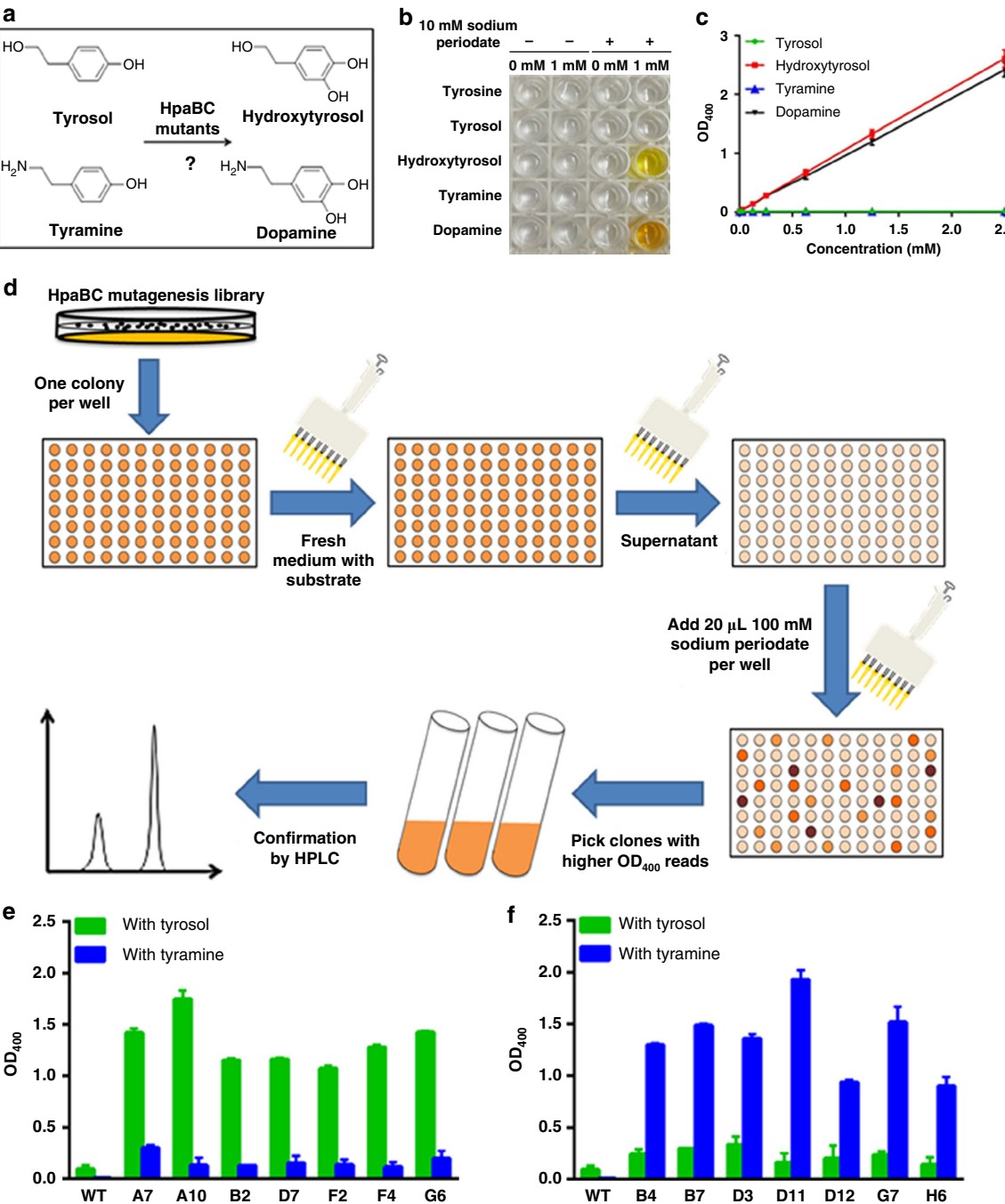

**Fig. 3** Directed divergent evolution of HpaBC. **a** Reactions targeted for directed divergent evolution; **b** Periodate assay of expected substrates and products of mutant HpaBCs. **c** Standard curves of the periodate assay; **d** Flowchart of the high-throughput screening of HpaBC; **e** OD$_{400}$ of the periodate assay of HpaBC mutants selected after positive screening with tyrosol and negative screening with tyramine; **f** OD$_{400}$ of the periodate assay of HpaBC mutants selected after positive screening with tyramine and negative screening with tyrosol. WT, wild-type. The data shown in **e** and **f** are from three replicate experiments and are expressed as the mean ± SD. Source data of **b**, **c**, **e** and **f** are provided in the Excel format Source Data file

accumulation was observed in this pathway, indicating that TYO activity on dopamine might be better than that on tyramine. Tyrosol accumulated in pathway 2.

**Construction and evaluation of a combined pathway.** Since intermediate accumulation occurred in each of the two pathways, we attempted to design a combined pathway in which pathways 1 and 2 proceeded simultaneously, with the aim to reduce the intermediate accumulation by dividing the biosynthetic carbon flow between the two pathways. Both steps catalyzed by tyrosol

hydroxylase and tyramine hydroxylase needed to be fulfilled in the combined pathway. Since protein overexpression is a major metabolic burden on engineered cells, a tyrosol/tyramine hydroxylase that exhibited promiscuous activities on both tyrosol and tyramine was developed to minimize the number of heterologous proteins overexpressed in the host cell. Thus, the same number of enzymes as in pathway 1 or 2 was used to accomplish the catalyses of the combined pathway (Fig. 6a).

HpaBC mutants active on both tyrosol and tyramine were again screened using the saturation mutagenesis library as

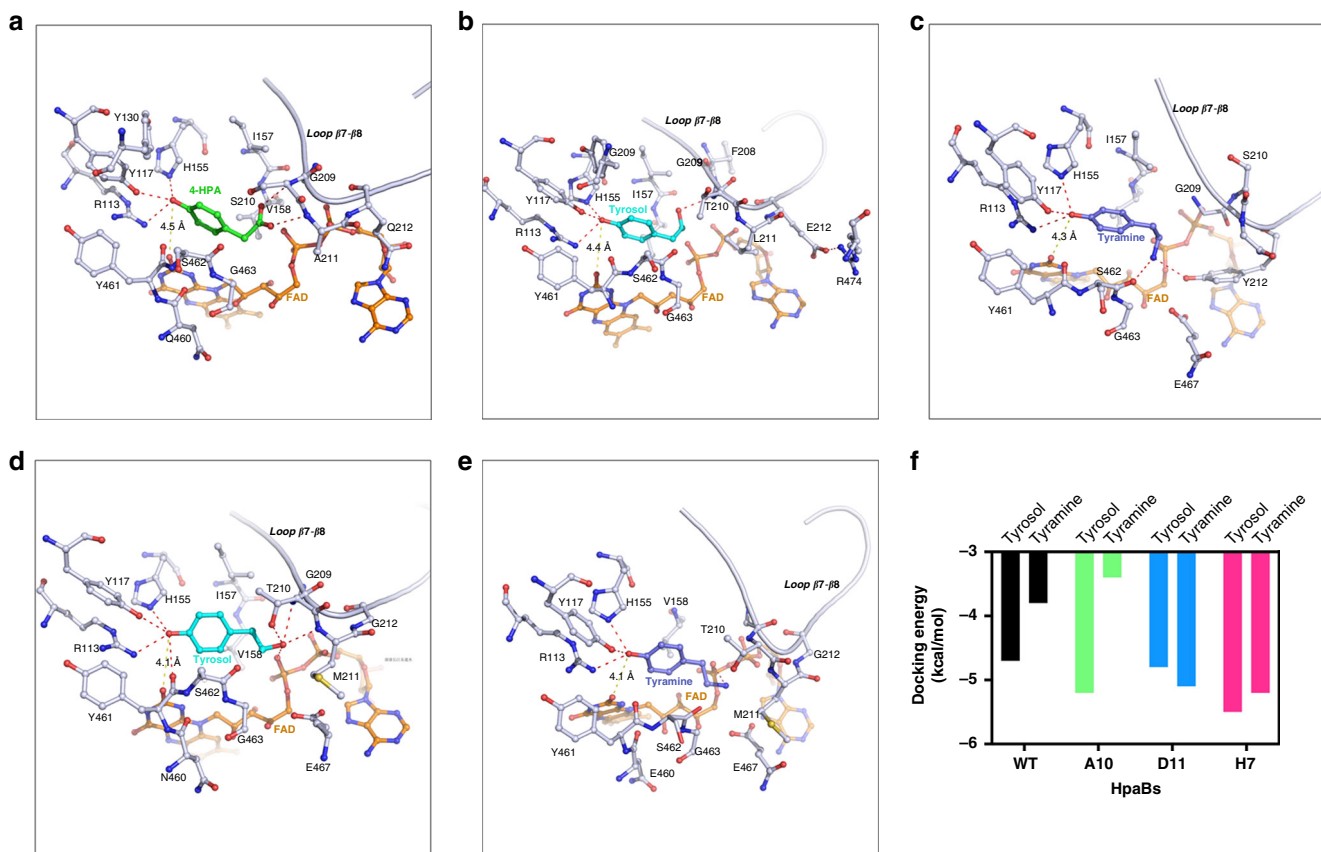

**Fig. 4** Molecular modeling of HpaB-substrate complexes. **a** Docking of 4-HPA to wild type HpaB, 4-HPA was shown as green stick; **b** Docking of tyrosol to mutant A10, tyrosol was shown as light blue stick; **c** Docking of tyramine to mutant D11, tyramine was shown as dark blue stick; **d** docking of tyrosol to mutant H7, tyrosol was shown as light blue stick; **e** Docking of tyramine to mutant H7, tyramine was shown as dark blue stick; FAD was colored as brown, hydrogen bonds were colored as magenta dash lines, and the distance from substrate to FAD were presented as yellow dash lines. **f** Docking energy of tyrosol and tyramine for HpaB wild type, A10, D11, and H7. The PDB files for generating **a–e** are provided as five independent Source Data files. In addition, source data to generate Fig. 4f are provided in the Excel format Source Data file

described above. The library was first screened for the highest activity in the presence of tyrosol and subsequently in the presence of tyramine. Seven mutants exhibiting significantly higher activity on both tyrosol and tyramine compared to the wild-type enzyme were obtained. The mutant H7 showed ~17- and ~271-fold higher activity on tyrosol and tyramine, respectively, as shown in Fig. 6b.

The mutant H7 was combined with TDC, TYO, and ADH to construct the combined pathway in which hydroxytyrosol was produced through both pathways 1 and 2. The time course of the biosynthesis was compared with those of pathway 1 and 2 (Fig. 6c). The hydroxytyrosol biosynthetic efficiency in the combined pathway was the highest by a considerable margin, whereas the accumulation of both tyrosol and tyramine was hardly detected. Hydroxytyrosol yield in the combined pathway reached 93%, which was much higher than that of pathway 1 (56%) or 2 (54%).

The results conclusively show that by taking advantage of promiscuous enzymes, the two biosynthetic pathways proceeded simultaneously and significantly elevated hydroxytyrosol biosynthetic efficiency. This is understandable because the carbon flow is divided between more than one pathway.

Early biosynthetic flux analysis was performed to explore the flux distribution between the two pathways, using tyramine as substrate. A colony of strain BHYT harboring pathway plasmid P6 carrying *hpaBC* gene encoding mutant H7, was grown in 3 mL M9Y medium for 12 h at 37 °C and diluted in 50 mL M9Y

medium with $OD_{600} = 0.1$, induced with 1 mM L-arabinose and supplemented with 15 mM tyramine. Samples were taken every hour from 1–6 h of culturing, and the concentrations of tyramine, dopamine, 3,4-DHPAA, 4HPAA, tyrosol, and hydroxytyrosol was quantified. The intermediate formation in this very early stage reflected the biosynthetic flux distribution between the two pathways (Fig. 6a). The results showed that tyrosol was the first intermediate to be detected from 1 h after inoculation. 4HPAA, the other intermediate in pathway 1, can be detected from 3 h. Nevertherless, dopamine and 3,4-DHPAA, the two intermediates in pathway 2, could not be detected until 6 h. At 6 h, dopamine accumulation was observed, and at the same time hydroxytyrosol showed up. However, 3,4-DHPAA was not detected all the time. The accumulations of 4HPAA and tyrosol disappeared from 6 h when hydroxytyrosol was observed (Table 1). It is possible that the H7 mutant became active enough after 5 h, simultaneously converting tyramine and tyrosol into dopamine and hydroxytyrosol, respectively, resulting in a rapid rise of hydroxytyrosol titer.

Substrate-feeding experiments were performed to synthesize hydroxytyrosol from tyrosine. A colony of strain BHYT harboring pathway plasmid P6 carrying *hpaBC* gene encoding mutant H7, was grown in 3 mL M9Y medium for 12 h at 37 °C, and diluted in 50 mL M9Y medium with $OD_{600} = 0.1$, induced with 1 mM L-arabinose. The substrate was supplemented in different ways, fed with 15 mM tyrosine when inoculation, fed with 7.5 mM tyrosine at 0 and 12 h after inoculation, or fed with 5 mM tyrosine at 0, 12, and 24 h after inoculation. Samples were

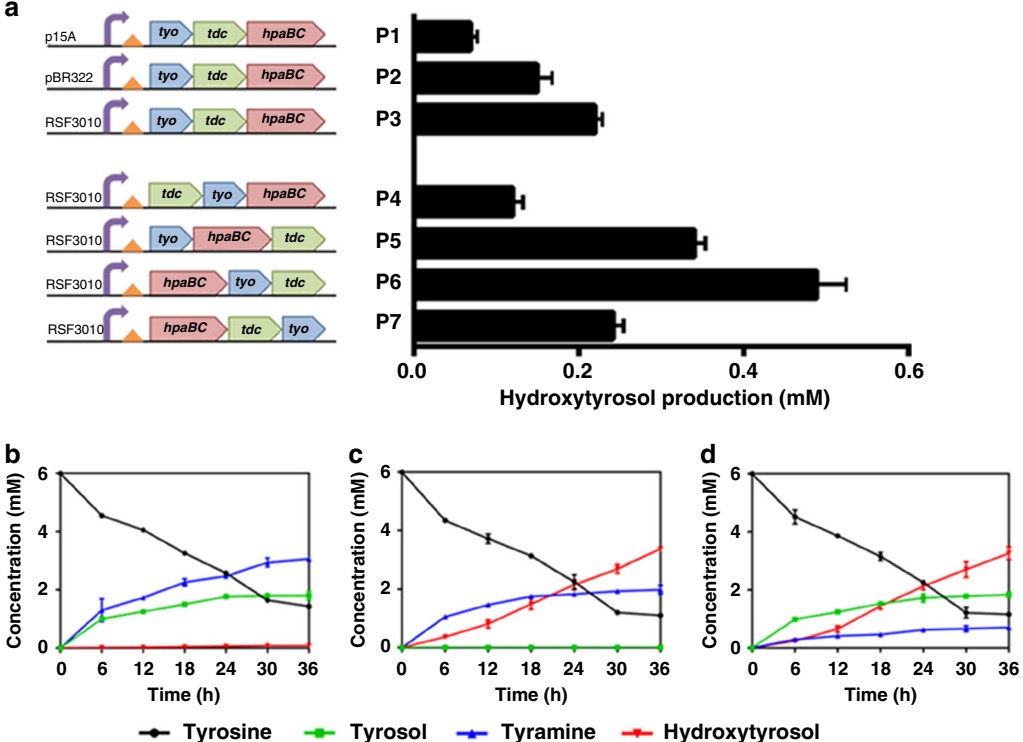

**Fig. 5** Hydroxytyrosol biosynthesis by pathways 1 and 2. **a** Optimization of plasmid copy number and gene order for improved hydroxytyrosol production. The time courses of product and intermediate production by the control pathway (containing wild-type HpaBC) (**b**), pathway 1 (**c**) and pathway 2 (**d**) were determined. The data shown are from three replicate experiments and are expressed as the mean ± SD. Source data of **a**–**d** are provided in the Excel format Source Data file

taken at different time points and the concentrations of hydroxytyrosol were analyzed by HPLC (Fig. 6d). The results showed that after culturing for 36 h at 37 °C, the third way for tyrosine supplementation resulted in higher hydroxytyrosol production (1890 mg L$^{-1}$), indicating the advantage of fed-batch fermentation.

**HpaBC engineering**. In this study, the monooxygenase HpaBC from *E. coli* B was engineered into a tyrosol hydroxylase, a tyramine hydroxylase and a tyrosol/tyramine hydroxylase. To confirm the activities of the mutants selected, the pFA plasmid expressing HpaBC mutant A10, D11, or H7 was isolated from the selected mutant strains and retransformed into strain BW25113. After a 36-h culture in the presence of 3 mM tyrosol or tyramine as a substrate, the production of hydroxytyrosol or dopamine, respectively, was determined using HPLC. As shown in Fig. 7a, the strain expressing wild-type HpaBC showed extremely low hydroxytyrosol and almost no dopamine production. The strain expressing the A10 mutant produced significantly higher amounts of hydroxytyrosol than of dopamine. On the contrary, strains expressing the D11 mutant mainly showed dopamine production but very low hydroxytyrosol production. The strain expressing the H7 mutant showed similar levels of hydroxytyrosol and dopamine production, which were 10% and 30% lower than those produced by strains expressing mutants A10 and D11, respectively. Sequencing results revealed the amino acid substitutions in the selected HpaB mutants (Table 2).

Figure 7b illustrates the predicted catalytic process of HpaBC. The flavin reductase component HpaC supplies FADH$_2$ for HpaB, which is the oxidase component. FADH$_2$ reacts with an oxygen molecule and the substrate. This, in turn, introduces a

hydroxyl group in the *ortho* position of the aromatic ring. The oxidized FAD is then recycled to HpaC for the next cycle[25]. The HpaB mutants A10, D11, and H7 were purified, in addition to wild-type HpaB and HpaC (Supplementary Figure 3). The specific activities of these enzymes on tyrosine, tyrosol, and tyramine, as well as the substrate of the wild-type enzyme, 4-HPA, were also determined. We found that mutants A10 and D11 showed good catalytic specificity towards tyrosol and tyramine, respectively, whereas mutant H7 exhibited considerable catalytic activity towards both substrates (Fig. 7c). Notably, the HpaBC mutants showed some catalytic activity towards 4-HPA, which was however lower than that of the wild-type enzyme. Mutant D11 showed high specificity towards tyramine, and its activities on tyrosol and 4-HPA were 8% and 13%, respectively, of that on tyramine. All mutants showed little activity on tyrosine.

To understand the substrate specificities of the HpaB mutants, molecular models were generated (Fig. 4). Mutant A10 displayed a high activity for tyrosol, and its docking energy for tyrosol was also found to be much lower than that for wild-type HpaB. The hydroxyl group at the tail of tyrosol forms a hydrogen bond with nitrogen atom on main chain of T210, and the side chain of T210 and L211 provide hydrophobic interactions to facilitate substrate binding. The negatively charged side chain of E212 is likely to make bond to side chain of R474, maintaining the loop conformation for tyrosol binding (Fig. 4b).

Mutant D11 displayed a high catalytic activity using tyramine as a substrate. Similar as wild-type HpaB, the hydroxyl head of tyramine is hold by three catalytic residues, which reserves the catalytic activity of HpaB. The distance from substrate to FAD is 4.3 Å. The amine group of tyramine was hydrogen bound with side chain of Y212 and main chain of S462 (Fig. 4c). The binding energy of tyrosol to mutant D11 is a little lower than that of

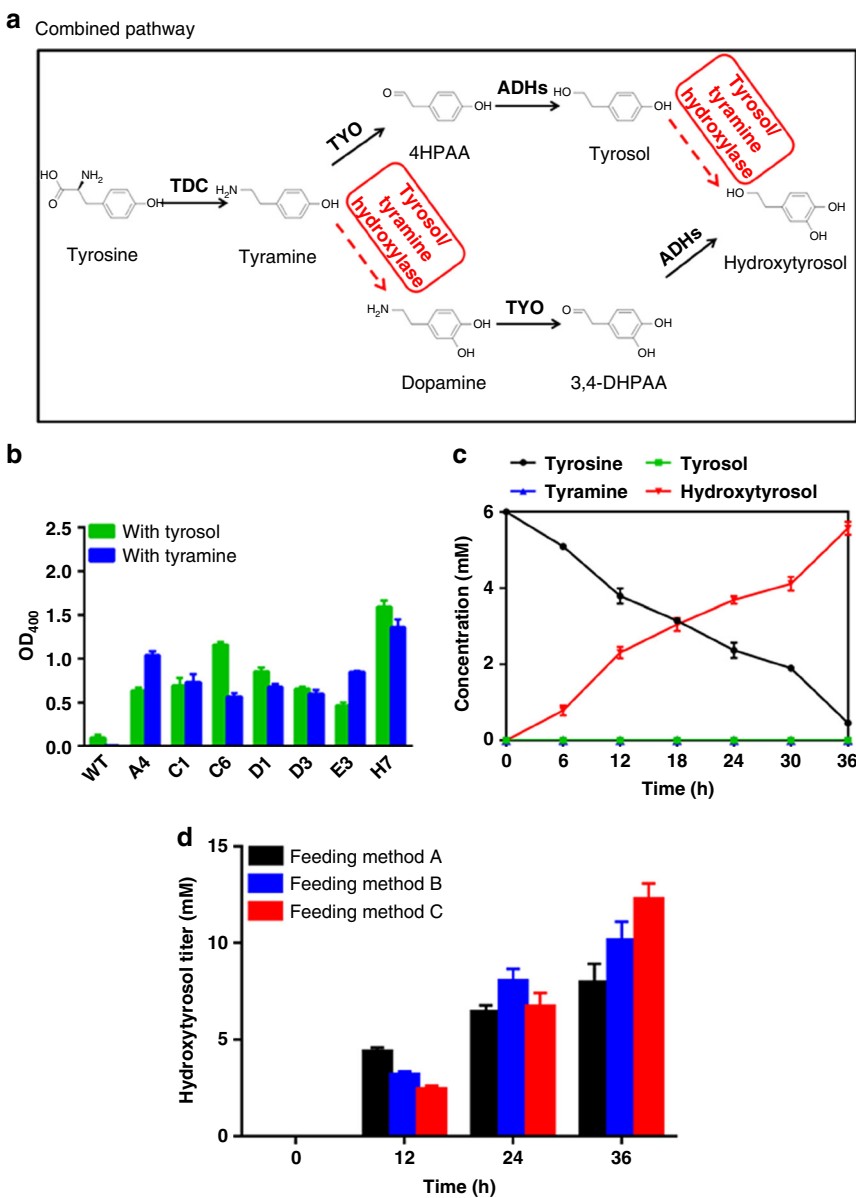

**Fig. 6** Construction of the combined pathway and its use for hydroxytyrosol biosynthesis. **a** The designed combined pathway. The dashed arrows indicate the steps lacking in highly efficient enzymes; **b** OD$_{400}$ of the periodate assay of HpaBC mutants selected after two rounds of positive screening with tyrosol and tyramine as substrates individually; WT, wild-type; **c** Time courses of product and intermediate production of the combined pathway; **d** Hydroxytyrosol productions under different substrate-feeding strategies. Feeding method A: fed with 15 mM tyrosine when inoculation; Feeding method B: fed with 7.5 mM tyrosine at 0 and 12 h after inoculation; Feeding method C: fed with 5 mM tyrosine at 0, 12, and 24 h after inoculation. The data are shown in **b**, **c** and **d** are from three replicate experiments and are expressed as the mean ± SD. Source data of **b**–**d** are provided in the Excel format Source Data file

**Table 1 Early flux analysis of the combined pathway, using tyramine as a substrate**

| Time (h) | 0 | 1 | 2 | 3 | 4 | 5 | 6 |
|---|---|---|---|---|---|---|---|
| Tyramine (mM) | 15.00 | 14.99 ± 0.0005 | 14.98 ± 0.0008 | 14.96 ± 0.0006 | 14.70 ± 0.05 | 14.40 ± 0.07 | 13.90 ± 0.008 |
| Dopamine (mM) | 0 | 0 | 0 | 0 | 0 | 0 | 0.05 ± 0.0008 |
| 3,4-DHPAA (mM) | 0 | 0 | 0 | 0 | 0 | 0 | 0 |
| 4HPAA (mM) | 0 | 0 | 0 | 0.10 ± 0.002 | 0.21 ± 0.009 | 0.12 ± 0.0004 | 0 |
| Tyrosol (mM) | 0 | 0.01 ± 0.0003 | 0.02 ± 0.0004 | 0.05 ± 0.0007 | 0.13 ± 0.002 | 0.52 ± 0.006 | 0 |
| Hydroxytyrosol (mM) | 0 | 0 | 0 | 0 | 0 | 0 | 1.12 ± 0.006 |

±Values represent standard deviations from three independent data points

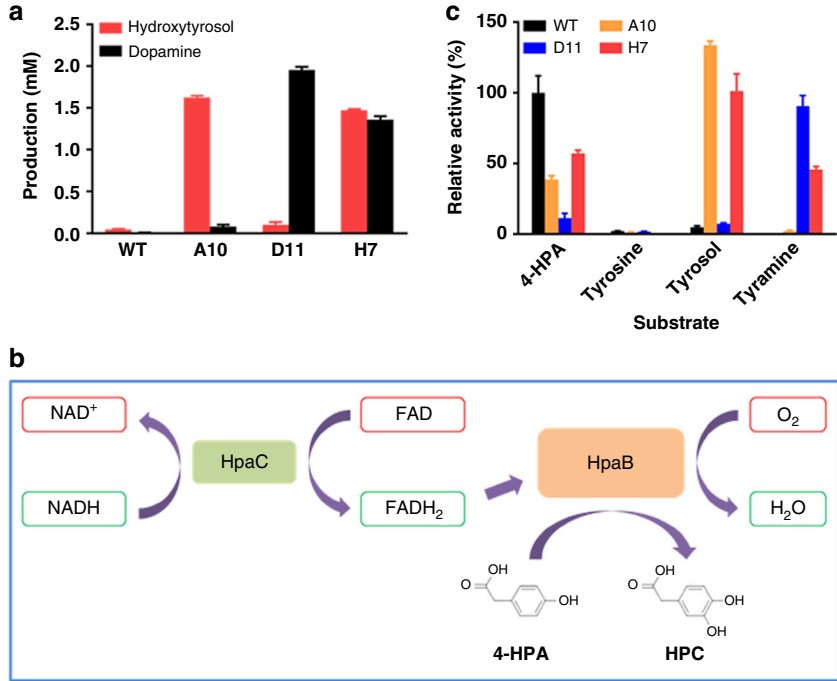

**Fig. 7** Characterization of wild-type HpaBC and mutants. **a** Hydroxytyrosol or dopamine produced by strain BW25113 expressing wild-type or the indicated mutant HpaBCs. Possible reaction scheme[25] (**b**) and specific activities of the wild-type and mutant HpaBCs on different substrates (**c**) are shown. The data are shown in **a** and **c** are from three replicate experiments and are expressed as the mean ± SD. Source data of **a**, **c** are provided in the Excel format Source Data file

**Table 2 The amino acid substitutions in HpaBC mutants**

| HpaBCs | 210 | 211 | 212 |
|---|---|---|---|
| Wild-type | S | A | Q |
| A10 | T | L | E |
| D11 | S | G | Y |
| H7 | T | M | G |

tyramine, and the hydroxyl group at the tail of tyrosol has ability to hydrogen bond with Y212 and S462. However, the hydroxyl group in the side chain of Y212 could be negatively charged in the physical condition, it will prefer the amine tail of tyramine which is likely to be positively charged. Introduction of tyrosine residues could be critical for recognition of tyramine, we also observed the tyrosine substitution at position 212 in mutant B7 (S210T, A211P, Q212Y) which preferred tyramine as a substrate (Fig. 3f).

Mutant H7 displayed a dramatically higher catalytic activity on both tyrosol and tyramine, and the docking energy is also found to decrease greatly compared with those for wild-type HpaB, suggesting a significant increase in affinity of H7 for both tyrosol and tyramine. Compared with A10, introduction of M211 and G212 in H7 is likely to increase flexibility of the loop, which will allow entrance of either tyrosol or tyramine. In the docking models, both tyrosol and tyramine are hydrogen bounded with oxygen atom from FAD (Fig. 4d, e). Moreover, hydroxyl of tyrosol can be also coordinated by side chain of T210, and main chains of G209 and M211 (Fig. 4d). Notably, the distance from substrate to FAD decreased to 4.1 Å, which may also enhance the catalytic efficiency of H7 for tyrosol and tyramine.

## Discussion
Diversity is necessary for evolutionary adaptation, which influences the survival of populations under environmental

variations[28–30]. For example, at the molecular level, the same reaction can be catalyzed by enzymes with totally different primary sequences[31,32]. On the contrary, an enzyme with promiscuous functions can catalyze different reactions[33–35]. This diversity enables various biosynthetic and metabolic processes in nature, leading to multiple possibilities for performing a physiological activity. In laboratory engineering, however, specificity is usually required. Under most circumstances, improved specific activity, or a balanced metabolic pathway with higher efficiency, is the focus of laboratory engineering[3,36]. For example, pathway enzyme activities are usually the bottleneck in obtaining high yields of natural products, and overexpression or activity engineering are the typical solutions[3,36]. In this study, mimicking nature, our study highlights the possibility of designing a multiple-pathway network consisting of pathways in which the catalytic steps all take place in alternative orders, such that the biosynthetic carbon flow is divided among multiple pathways and converts the substrate and intermediates to the final product to the maximum extent. As shown by the time-course results (Figs. 5, 6), the intermediate accumulation in pathways 1 and 2 was minimized in the combined pathway, leading to a significantly improved hydroxytyrosol yield. The design of a combined pathway is also a good choice to prevent possible feedback inhibition in pathway construction. Since overexpression of too many pathway enzymes usually imposes a metabolic burden on cells[37], which subsequently has negative effects on biosynthesis, promiscuous enzymes are good choices to minimize the number of enzymes overexpressed in the multiple-pathway network. Promiscuous enzymes can be obtained through protein engineering strategies, for example, we described the development of the H7 mutant by engineering the HpaBC enzyme.

In nature, it is believed that proteins with promiscuous activities acquire increased genetic diversity through gene duplication and mutation and evolve to diverging new proteins[38,39]. In the development of the tyrosol hydroxylase, tyramine hydroxylase

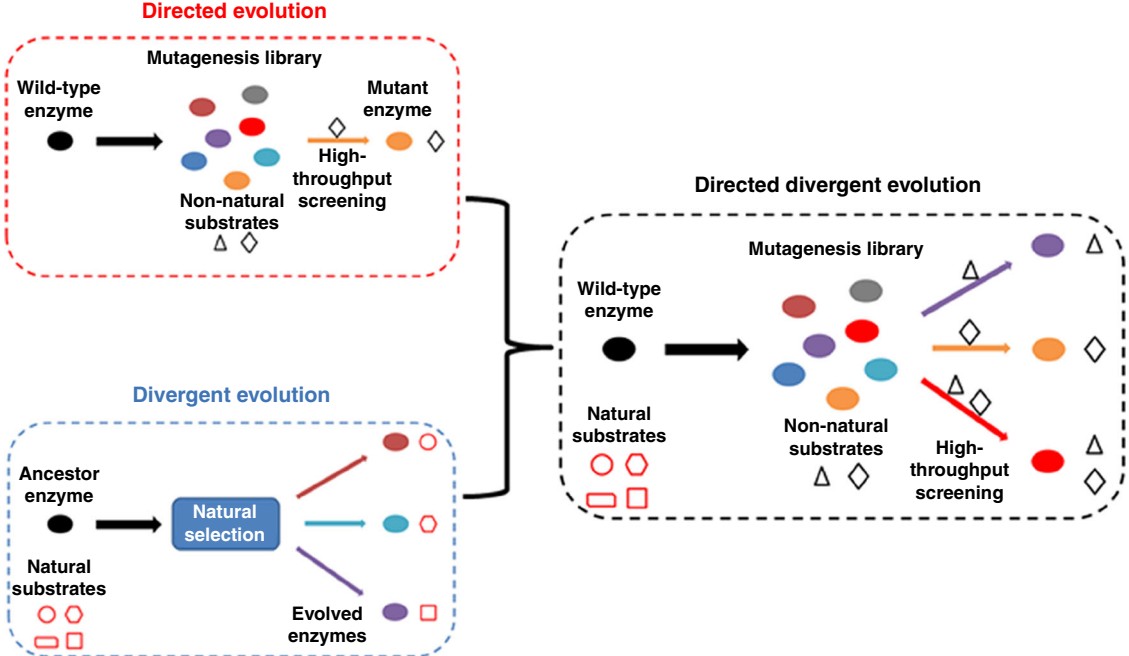

**Fig. 8** Scheme of directed divergent evolution. A parental enzyme can be evolved to develop multiple enzymes with altered substrate specificity. Directed evolution is undertaken and high-throughput screening is used to select the desired mutant enzymes

and tyrosol/tyramine hydroxylase by engineering of HpaBC, the mechanisms of divergent evolution and directed evolution were combined. The three mutant enzymes, A10, D11, and H7, showed different substrate specificities. For substrate tyramine, the wild-type HpaBC hardly exhibited any activity. We called this strategy directed divergent evolution, with which a parental enzyme can be evolved to develop multiple enzymes with altered substrate specificity (Fig. 8). It can be characterized as follows: (i) the parental enzyme may or may not be a promiscuous enzyme; (ii) the evolved enzymes are highly specific or promiscuous for other substrates that are not accepted by the parental enzyme; (iii) directed evolution is undertaken and high-throughput screening is used to select the desired mutant enzymes. Moreover, negative screening can be applied to weed out undesired activities. Directed divergent evolution is a powerful tool in introducing different substrate specificity for biosynthetic intermediates into corresponding pathway enzymes. By replacing the pathway enzymes with their promiscuous mutant counterparts, a multiple-pathway network can be constructed.

Using directed divergent evolution, we have shown the possibility of developing mutant enzymes specifically accepting or rejecting a particular compound as a substrate by positive or negative screening, respectively, irrespective of whether the wild-type enzyme showed the same activity or not. Therefore, the directed divergent evolution strategy has the potential to develop mutant enzymes with divergent activities from the same parental enzyme. For 4-HPA, which was not subjected to negative screening, the activity of mutants A10 and D11 was ~39% and ~11%, respectively, of that observed for their own optimal substrates. The H7 mutant, active on both tyrosol and tyramine, also showed activity on 4-HPA similar to that on tyramine (Fig. 7c). Thus, the development of different activity or improvement of promiscuous activity does not seem to trade-off with parallel decreases in the original activity. However, removal of a particular activity can be achieved by introducing the negative screening, such as the development of mutant A10 or D11, showing negative activity on tyramine or tyrosol, respectively.

Hydroxytyrosol productions using wild-type HpaBC have been reported[15,16,40,41]. High-level HpaBC expression using strong promoter and high-copy plasmid was used to promote the conversion[40,41], otherwise tyrosol accumulations occurred[15,16], indicating the necessity to optimize HpaBC activity on tyrosol. In this study, the expression level of HpaBC, as well as other pathway enzymes, was kept low, in order to minimize the metabolic burden of the cells. Further improved HpaBC activity on tyrosol significantly elevated the hydroxytyrosol production under the minimized protein expression burden. Low yeast extract content in the medium could also lead to the low hydroxytyrosol production using wild-type HpaBC in this study, compared with the previous reports[15,16,40,41]. So far the highest hydroxytyrosol production from tyrosine was reported by Li et al.[15], with a different hydroxytyrosol biosynthetic pathway consisting of TyrB, ketoacid decarboxylase (KDC), ADH, and wild-type HpaBC. The hydroxytyrosol production was 1243 mg L$^{-1}$ (~48% yield), with the combination of removing $NH_4Cl$ and adding 1-dodecanol and ascorbic acid in the medium which led to 75% improved production. In our study, the hydroxytyrosol production reached 1890 mg L$^{-1}$ (~82% yield), under no medium optimization.

In conclusion, the directed divergent evolution strategy is a powerful tool in developing various catalytic activities for biosynthetic purposes. In this study, we demonstrate the successful application of the HpaBC mutants developed by directed divergent evolution in constructing hydroxytyrosol biosynthetic pathways. The idea of designing multiple-pathway networks based on promiscuous pathway enzyme activities, in which the substrate and intermediates are efficiently directed to the product and the metabolic burden is minimized, is of great significance in metabolic engineering studies.

## Methods
**General**. Restriction enzymes, DNA polymerase, T4 polynucleotide kinase, and T4 DNA ligase were purchased from Takara Bio, Inc. (Dalian, China). Hieff Clone™ Plus Multi One Step Cloning Kit was purchased from YEASEN Biotechnology Co.,

Ltd. (Shanghai, China). Oligonucleotides were purchased from Life Technologies (Shanghai, China). 4-Hydroxyphenylacetate, tyrosine, tyrosol, hydroxytyrosol, tyramine, dopamine, 3,4-DHPAA, 4HPAA, sodium periodate, dansylcholoride, and NADH were all purchased from Sigma-Aldrich (St. Louis, USA).

*Escherichia coli* strain MC1061, BL21(DE3), and BW25113 were used for cloning, protein overexpression, and hydroxytyrosol biosynthesis, respectively. Strains were grown at 37 °C. Kanamycin (50 μg/mL) were supplemented when necessary. Hydroxytyrosol biosynthesis was performed in yeast extract M9Y medium (M9 minimal salts (Becton, Dickinson and Company), 1% (w/v) glucose, 5 mM MgSO$_4$, 0.1 mM CaCl$_2$ supplemented with 0.025% (w/v) of yeast extract)[14], while other cultures were done in Luria-Bertani (LB) medium if not specially indicated.

**Strain construction**. The FRT-flanked *kan* gene was eliminated from strain JW1380-KC (Keio collection)[42], using flipase-mediated recombination[43]. Strain JW1380-KC was transformed with plasmid pCP20, and ampicillin-resistant transformants were selected at 30 °C, after which a few were colony-purified once nonselectively at 42 °C and then tested for loss of all antibiotic resistances. The majority lost the *kan* resistance gene and the FLP helper plasmid simultaneously, resulting in strain BHYT in which the *feaB* gene was deleted. The deletion was verified by PCR.

**Plasmid construction**. All strains, plasmids and primers used in this study are listed in Supplementary Tables 1 and 2, respectively. The replication origin of plasmid pBAD18-Kan[44] was replaced with RSF3010 and p15A origins, resulting in plasmid pRSF and pFA, respectively. The *hpaBC* gene encoding a monooxygenase (Genbank accession No. CP020368.1) was amplified using the genomic DNA of strain BL21 (DE3) as template with primers *hpaB*-for-*Nde*I and *hpaC*-rev-*Xho*I. Then after digestion with *Nde*I and *Xho*I, the purified *hpaBC* gene fragment was ligated into the PCR product amplified with plasmid pFA as template using primers pFA-for-*Xho*I and pFA-rev-*Nde*I, resulting in pFA-*hpaBC*.

The *tdc* gene encoding tyrosine decarboxylase (Genbank accession No. CP014949.1) was amplified from the genome of strain *Enterococcus faecalis*. The *tyo* gene from *Micrococcus luteus* encoding tyramine oxidase (Genbank accession No. AB010716.1) was synthesized by General Biosystems (Chuzhou, China) after codon optimization (Supplementary Table 3). Purified gene fragments *tdc*, *tyo* and *hpaBC* were assembled with the vector fragment of pBAD18-Kan, pRSF or pFA, using Hieff Clone Plus Multi One Step Cloning Kit, resulting in plasmids pFA-*tyo*-*tdc*-*hpaBC* (P1), pBAD18-*tyo*-*tdc*-*hpaBC* (P2), pRSF-*tyo*-*tdc*-*hpaBC* (P3), pRSF-*tdc*-*tyo*-*hpaBC* (P4), pRSF-*tyo*-*hpaBC*-*tdc* (P5), pRSF-*hpaBC*-*tyo*-*tdc* (P6) and pRSF-*hpaBC*-*tdc*-*tyo* (P7). The three genes in different orders were organized into an operon expressed from the above plasmids.

The wild-type or mutant *hpaB* genes were amplified with primers *hpaB*-for-*Nde*I and *hpaB*-rev-*Xho*I using plasmid pFA-*hpaBC* as template, and ligated into plasmid pET28a after digestion with *Nde*I and *Xho*I, resulting in plasmid pET28a-*hpaB* carrying wild-type or mutant *hpaB* gene. The *hpaC* gene was amplified with primers *hpaC*-for-*Nde*I and *hpaC*-rev-*Xho*I using plasmid pFA-*hpaBC* as template, and ligated into plasmid pET28a after digestion with *Nde*I and *Xho*I, resulting in plasmid pET28a-*hpaC*.

**HpaBC library construction**. A site-saturation mutagenesis library of HpaBC with residues S210, A211, and Q212 in the HpaB component randomized was constructed. PCR was performed with primers *hpaBC*-Saturated-fwd and *hpaBC*-Saturated-rev using pFA-*hpaBC* as template, and the PCR product was used as a megaprimer to perform MEGAWHOP PCR[45] using pFA-*hpaBC* as template. *Dpn*I (20 U) digestion was performed at 37 °C for 2 h and then inactivated at 80 °C for 20 min. Then the PCR products were used to transform strain *E. coli* MC1061 and around 10$^5$ transformants were recovered. Randomly picked clones from this library were sequenced, and mutations at the indicated positions were confirmed, with no additional point mutations.

**High-throughput screening of the mutagenesis library**. Single colonies of strain BW25113 harboring mutants from the HpaBC site-saturation mutagenesis library were grown in 0.8 mL M9Y medium in 96-well plates at 30 °C for 24 h, induced with 1 mM L-arabinose and supplemented with 6 mM tyrosol or tyramine as substrate. Cells were harvested by centrifugation (1840 × *g*, 10 min, 4 °C). 180 μl of the supernatant was added into 20 μl of 100 mM sodium periodate, then OD$_{400}$ of the reaction mixture was determined with a SynergyMx Multi-Mode Microplate Reader (BioTek, Vermont, USA). Mutants showing highest and lowest absorbance in positive and negative screening, respectively, were selected. The OD$_{400}$ and product formations of the selected mutants were confirmed again in test-tube cultures.

**HPLC quantification**. A colony of strain BW25113 harboring plasmid pFA-*hpaBC* or strain BHYT harboring pathway plasmid, carrying wild-type or mutant *hpaBC* gene, was grown in M9Y medium at 37 °C till OD$_{600}$ = 0.6, then induced with 1 mM L-arabinose. After further grown at 37 °C for 24 h, the cell culture was centrifuged at 12,000 × *g* for 5 min, and the supernatant was filtrated through a 0.22 μm filter and analyzed with HPLC.

Concentrations of 4-HPA, tyrosine, 3,4-DHPAA, 4HPAA, tyrosol, and hydroxytyrosol were determined by HPLC[46]. HPLC was performed using Shimadzu LC-20A system (Shimadzu Corp., Kyoto, Japan) equipped with a SPD-M20A photodiode array (PDA) detector operating at 280 nm. Separation was achieved using a Waters Symmetry C18 column (250 × 4.6 mm, 5 μm) (Waters, USA) working at 30 °C. Mobile phase A was 5 % acetic acid, and B was acetonitrile. A linear gradient of mobile phase B (0–20 min, increased from 15 to 40%; 20–25 min, increased from 40 to 100%) with a flow rate of 0.8 mL/min was used for separation. The concentrations were calculated from standard curves prepared with corresponding authentic compounds.

The concentrations of tyramine and dopamine were determined after derivation with dansylcholoride[47]. HPLC was performed using Shimadzu LC-20A system equipped with a SPD-M20A PDA detector operating at 254 nm. Separation was achieved using a Waters Symmetry C18 column (250 × 4.6 mm, 5 μm) working at 30 °C. Mobile phase A was water and B was acetonitrile. A linear gradient of mobile phase B (0–7 min, increased from 45 to 60%; 7–25 min, increased from 60 to 90%; 25–40 min, 90%) with a flow rate of 0.8 mL/min was used for separation. The concentrations were calculated from the standard curves prepared with corresponding authentic compounds.

**Protein purification**. A single colony of strain BL21 (DE3) harboring plasmid pET28a-*hpaC* or pET28a-*hpaB* carrying wild-type or mutant *hpaB*, was grown in LB medium at 37 °C till OD$_{600}$ = 0.6, then induced with 0.4 mM IPTG. After further grown at 30 °C for 12 h, cells were collected by centrifugation at 4 °C, resuspended in lysis buffer (50 mM Tris-HCl, 300 mM NaCl, 10 mM imidazole, pH 8.0) and disrupted by sonication with a JY92-IIN Ultra Sonic Cell Crusher (Ningbo, China). After centrifugation, the supernatant was loaded on a pre-equilibrated nickel-nitrilotriacetic acid (Ni-NTA) column (Qiagen, Valencia, USA). The column was washed with washing buffer (50 mM Tris-HCl, 300 mM NaCl, 20 mM imidazole, pH 8.0), and the bound protein was then eluted with elution buffer (50 mM Tris-HCl, 300 mM NaCl, 250 mM imidazole, pH 8.0). Imidazole was removed by dialysis at 4 °C against 50 mM Tris-HCl buffer (pH 8.0). The purity of proteins was assessed by sodium dodecyl sulfate-polyacrylamide gel electrophoresis (SDS-PAGE) and the protein concentration was assayed with Bradford method[48].

**Activity assay of HpaBC**. The HpaC activity was assayed at 37 °C by monitoring the absorbance decrease at 340 nm which was corresponding to the concentration variation of NADH, in the absence of HpaB[49].

For HpaB activity determination, the reaction mixture containing 0.1 mM FAD, 1 mM NADH, 0.1 mL of purified HpaC and 0.5 mL of purified HpaB wild-type or mutant enzyme in 50 mM Tris-HCl buffer (pH 8.0) was incubated at 37 °C. The reaction was started by adding 5 mM of the indicated substrate (4-HPA, tyrosine, tyrosol or tyramine), and stopped with 5% (v/v) acetic acid after 1 h. The reaction mixture was then centrifuged at 12,000 × *g* for 10 min, and the supernatant was quantified by HPLC. One unit of HpaB activity was defined as the amount of enzyme catalyzing the conversion of 1 mM of the corresponding substrate per minute under the conditions described above[49].

**Homology modeling**. The three dimension structure of HpaB of *E. coli* was generated by Modeler 9.20 using HpaB from (4OO2) and (2YYJ) as templates[24,25,50]. Mutants were manually substituted native amino acid with desired sequences in Coot[51], and followed with a loop refinement by Modeller 9.20 due to its high flexibility. Protein structure generated from each model process for wild type and mutants were submitted to SAVE v5.0 (http://servicesn.mbi.ucla.edu/SAVES/) server for assessment, and the model with the highest overall quality factor and Ramachandran plot were chosen as final models. The final models were further optimized by energy minimization with Gromacs 96 force field in Gromacs-5.0.7[52]. The binding model of 4-HPA, tyrosol, tyramine to wild type or mutant HpaBs were calculated by Autodock Vina[53], binding energies were used to estimate the affinities of substrates.

**Reporting summary**. Further information on experimental design is available in the Nature Research Reporting Summary linked to this article.

## Data availability

The datasets generated and analyzed during the current study are available from the corresponding author on reasonable request. A reporting summary for this article is available as a Supplementary Information file. The source data underlying Figs. 3b, c, e, f, 4f, 5a–d, 6b–d and 7a, c, Supplementary Fig. 3A, B can be found in the Excel format Source Data file. In addition, the PDB files for generating Fig. 4A–E are provided as five independent Source Data files.

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

## Acknowledgements

This work was supported by the National Natural Science Foundation of China (Grant No. 31501037, 31870072, and 21472234).

## Author contributions

S.-Y.T., J.-M.J., Y.H. and W.C. conceived the idea and supervised the research. W.C. and J.Y. did the construction of strains, plasmids, and library. J.Y. did the screening of mutants. W.C., J.Y., Y.G. and J.M. did the HPLC analysis. W.C., J.Y. and W.H. did protein purification and enzyme activity assay. Y.H. did homology modeling, docking, and related analysis. W.C. did feeding experiment and early flux analysis. Y.T. contributed reagents/ materials/analysis tools. Y.T., Y.C., Y.G. and G.S. critically reviewed the manuscript. S.-Y.T., J.-M.J., Y.H. and W.C. performed data analysis and wrote the paper. All of the authors reviewed, approved and contributed to the final version of the manuscript.

## Additional information

**Competing interests:** The authors declare no competing interests.

