## [Peer Review File · Nature Communications]

Journal peer review information: *Nature Communications* thanks Toshiki Furuya, Pablo Carbonel, and the other anonymous reviewer(s) for their contribution to the peer review of this work. Peer reviewer reports are available.

Reviewers' comments:

Reviewer #1 (Remarks to the Author):

In nature, some enzymes catalyze multiple reactions due to their promiscuous activities. Based on this principle, Chen et al. designed dual hydroxytyrosol biosynthetic pathway by employment of monooxygenase HpaB. The wild type HpaB shows low activity towards two substrates (tyrosol and tyramine). In order to get highly efficient HpaB, three amino acids in HpaB binding sites were chosen for simultaneous saturation mutagenesis. After high-throughput screening, mutant A10 and D11 showed highest activity towards tyrosol and tyramine, exhibiting 19-fold and 386-fold higher activity on their corresponding substrates than the wild-type enzymes, respectively. Mutant H7 exhibited higher activity on both tyrosol (17-fold) and tyramine (271-fold) compared to the wild-type enzyme. This study demonstrates a strategy that the pathway efficiency can be improved by rearranging the metabolic flux among multiple pathways by using promiscuous enzymes.

However, this research is based on the work of Li and used the same key enzymes. Compared to the previous study, the experiments and results of this manuscript are not that significant. In this manuscript, the key work is saturation mutagenesis of HpaB, which has been broadly used in protein engineering. Taken together, this manuscript seems to lack the novelty to guarantee the publication in the journal. For those reasons, it is suggested that the authors perform additional manipulations and re-submit.

Reviewer #2 (Remarks to the Author):

This manuscript describes biosynthesis of hydroxytyrosol from tyrosine using *Escherichia coli* cells expressing mutants of a flavin-dependent monooxygenase, HpaBC. Biosynthesis of hydroxytyrosol using *E. coli* cells expressing HpaBC has already been reported by at least two groups (Li X, et al., *ACS Synth Biol*, 7, 647 (2018); Chung D, et al., *Sci Rep*, 7, 2578 (2017)). However, in this manuscript, the authors did not explain the results reported in these previous papers. The authors suggested a new concept, directed divergent evolution. Although the idea is interesting, it does not include innovative technique.

1. Fig. 4 and Fig. 6

Biosynthesis of hydroxytyrosol from tyrosol using HpaBC has already been reported, as described above. In these previous reports, hydroxytyrosol was synthesized from tyrosol using the wild-type HpaBC. In contrast, the wild-type HpaBC used by the authors did not show activity for tyrosol. Please discuss the different results. Also, please show the production yield of hydroxytyrosol (mg/L) and discuss it in comparison with the production yield attained by others.

2. P.4, L108

Although the authors described that "the activity was not sufficient due to its low substrate specificity on tyrosol", activity is generally not related to substrate specificity.

3. P.8, L221

The authors altered the substrate specificity of HpaBC by site-directed mutagenesis. Please discuss why the mutants show activity for tyrosol and/or tyramine from the viewpoint of amino acid substitution.

4. Fig. 3D

"Fresh medium with substrate" and "Supernatant" should be written under the arrows.

Reviewer #3 (Remarks to the Author):

In the manuscript from Chen et al, the authors engineered a promiscuous hydroxylase from wild type HpaBC acting on both tyrosol and tyramine substrates using a "directed divergent evolution" strategy. For that purpose, the authors developed a screening technique based on colorimetric differences. The resulting mutants were used to activate simultaneously two hydroxytyrosol biosynthetic pathways, lowering in that way the metabolic burden by flux redistribution.

Using saturation mutagenesis based on the HpaBC crystal structure from *Thermus Thermophilus* HB8, mutants were developed showing significantly higher activity either on tyrosol or tyramine, but not on both. Similarly, mutants that were active on both tyrosol and tyramine were screened.

The novelty is the use of directed evolution as a way of developing a mutant with promiscuous activity towards two substrates that can be simultaneously active in two alternative biosynthetic pathways.

The work presents convincing results. However, it is not completely clear how fluxes were distributed for mutant H7 (the highest producer with activity for both tyrosol and tyramine). Results shown in Figure 5C (time course) indicate no accumulation of intermediates, which was not previously observed in other mutants. Is there a way of investigating further how fluxes were distributed in the pathways?

All issues are carefully addressed as follows and implemented in the revised manuscript. The changes were indicated in yellow in the revised manuscript.

Reviewer #1 (Remarks to the Author):

In nature, some enzymes catalyze multiple reactions due to their promiscuous activities. Based on this principle, Chen et al. designed dual hydroxytyrosol biosynthetic pathway by employment of monooxygenase HpaB. The wild type HpaB shows low activity towards two substrates (tyrosol and tyramine). In order to get highly efficient HpaB, three amino acids in HpaB binding sites were chosen for simultaneous saturation mutagenesis. After high-throughput screening, mutant A10 and D11 showed highest activity towards tyrosol and tyramine, exhibiting 19-fold and 386-fold higher activity on their corresponding substrates than the wild-type enzymes, respectively. Mutant H7 exhibited higher activity on both tyrosol (17-fold) and tyramine (271-fold) compared to the wild-type enzyme. This study demonstrates a strategy that the pathway efficiency can be improved by rearranging the metabolic flux among multiple pathways by using promiscuous enzymes.

However, this research is based on the work of Li and used the same key enzymes. Compared to the previous study, the experiments and results of this manuscript are not that significant. In this manuscript, the key work is saturation mutagenesis of HpaB, which has been broadly used in protein engineering. Taken together, this manuscript seems to lack the novelty to guarantee the publication in the journal. For those reasons, it is suggested that the authors perform additional manipulations and re-submit.

Thank you for the suggestions.

In the work of Li (Li et al. ACS Synthetic Biology, 2018, 7:647), the pathway included TyrB, ketoacid decarboxylase (KDC), ADH and HpaBC. While in our study, the designed pathway included enzymes TDC, TYO, ADH and HpaBC. **The key enzymes of these two pathways are different**, and the first two catalytic steps are

also different.

About the novelty, in our work, we expected to put forward the idea of designing multiple-pathway networks based on promiscuous pathway enzyme activities, in which the biosynthetic carbon flow was divided in multiple pathways, efficiently directing substrate and intermediates to the product. At the same time, the metabolic burden is minimized, since the same number of enzymes was used as in single-pathway biosynthesis. The strategy used in this study was different from the regularly used strategies such as overexpressing or engineering key enzymes, balancing expressions of pathway enzymes, removing branch pathways, and so on. This strategy is potential to be widely used in metabolic engineering. To use this strategy, promiscuous pathway enzymes are needed. We demonstrated the possibility to develop promiscuous enzymes via directed evolution. From the same parental enzyme, mutants with different substrate specificities were evolved, no matter the parental enzyme accepts the substrates or not.

Reviewer #2 (Remarks to the Author):

This manuscript describes biosynthesis of hydroxytyrosol from tyrosine using Escherichia coli cells expressing mutants of a flavin-dependent monooxygenase, HpaBC. Biosynthesis of hydroxytyrosol using E. coli cells expressing HpaBC has already been reported by at least two groups (Li X, et al., ACS Synth Biol, 7, 647 (2018); Chung D, et al., Sci Rep, 7, 2578 (2017)). However, in this manuscript, the authors did not explain the results reported in these previous papers. The authors suggested a new concept, directed divergent evolution. Although the idea is interesting, it does not include innovative technique.

1. Fig. 4 and Fig. 6

Biosynthesis of hydroxytyrosol from tyrosol using HpaBC has already been reported, as described above. In these previous reports, hydroxytyrosol was synthesized from tyrosol using the wild-type HpaBC. In contrast, the wild-type HpaBC used by the authors did not show activity for tyrosol. Please discuss the different results. Also, please show the production yield of hydroxytyrosol (mg/L)

and discuss it in comparison with the production yield attained by others.

Thank you for the suggestions

We have shown that wild-type HpaBC **was active** on tyrosol, but the activity was as low as 4.87% of that on the optimal substrate 4-HPA (Fig. 6C). The substrate for which HpaBC showed little activity is tyramine. In this work we improved HpaBC activity on tyrosol and evolved a new substrate specificity for tyramine.

Since in our study the expression level of HpaBC was low, the hydroxytyrosol production was as low as 12.33 mg/L using tyrosine as substrate when wild-type HpaBC was used, and tyrosol was accumulated (Fig. 5B). But low pathway expression would reduce the metabolic burden of the cells, which benefit the product formation, as shown in Fig. 5A. Further improved activity of HpaBC on tyrosol significantly elevated the hydroxytyrosol production when pathway overexpression burden was minimized. Another possible reason for the low hydroxytyrosol production using wild-type HpaBC is that only 0.25 g/L yeast extract was used in medium in our study, compared with 1-2 g/L of yeast extract supplementation in other reports (Lieb Gott et al. *Research in Microbiology* 2009, 160:757; Li et al. *ACS Synth. Biol.* 2018,7:647; Chung et al. *Scientific Reports* 2017, 7:2578; Choo et al. *Appl. Biol. Chem.* 2018, 61:295). Yeast extract may improve the HpaBC expression and it also contains tyrosine. In some reports, when wild-type HpaBC was used to produce hydroxytyrosol, tyrosol or other intermediates also accumulated (Lieb Gott et al. *Research in Microbiology* 2009, 160:757; Li et al. *ACS Synth. Biol.* 2018,7:647), indicating the necessity to optimize HpaBC activity on tyrosol. In some other reports, the HpaBC was expressed via strong promoter (T7) on high-copy plasmid, so that the enzyme was expressed in high levels (Chung et al. *Sci. Rep.* 2017, 7:2578; Choo et al. *Appl. Biol. Chem.* 2018, 61:295), to efficiently converting tyrosol to hydroxytyrosol. We have discussed this in the revised manuscript (page 13, paragraph 3).

Hydroxytyrosol can be synthesized from tyrosol, tyrosine or glucose. To better compare our hydroxytyrosol productions from tyrosine with previous reports, substrate-feeding experiments were performed (Figure 6D). The results showed that

when 5 mM tyrosine was fed at 0, 12 and 24 h after inoculation, hydroxytyrosol production reached **1890 mg/L** after culturing for 36 h at 37°C, with a yield of 82%. So far the highest hydroxytyrosol production from tyrosine was reported by Li et al. (ACS Synth. Biol. 2018, 7:647), via a different hydroxytyrosol biosynthetic pathway consisting of TyrB, KDC, ADH and wild-type HpaBC. The hydroxytyrosol production at 36 h was 1243 mg/L (8.06 mM) when 16.56 mM tyrosine was added (fed at 0, 12, 24 h after inoculation, each time at 5.5 mM (~48% yield), with the combination of removing NH₄Cl and adding 1-dodecanol and ascorbic acid in the medium which led to 75% improved production. For the hydroxytyrosol production in this study, no medium optimization was performed. We have supplemented the result in page 9 (paragraph 2), page 14 (paragraph 1) and Figure 6D.

2. P.4, L108

Although the authors described that "the activity was not sufficient due to its low substrate specificity on tyrosol", activity is generally not related to substrate specificity.

Sorry, we did not explain it clearly enough. We have corrected it as "the activity was not sufficient, as tyrosol is not the optimal substrate of wild-type HpaBC and this tyrosol hydroxylation activity was relatively low." (page 4, paragraph 3)

3. P.8, L221

The authors altered the substrate specificity of HpaBC by site-directed mutagenesis. Please discuss why the mutants show activity for tyrosol and/or tyramine from the viewpoint of amino acid substitution.

Thank you for the suggestion. To explore this, molecular models of these HpaB variants were generated (Figure 4).

Mutant A10 displayed a high activity for tyrosol, and its docking energy for tyrosol was also found to be much lower than that for wild-type HpaB. The hydroxyl group at

the tail of tyrosol forms a hydrogen bond with nitrogen atom on main chain of T210, and the side chain of T210 and L211 provide hydrophobic interactions to facilitate substrate binding. The negatively charged side chain of E212 is likely to make bond to side chain of R474, maintaining the loop conformation for tyrosol binding (Figure 4B).

Mutant D11 displayed a high catalytic activity using tyramine as a substrate. Similar as wild-type HpaB, the hydroxyl head of tyramine is hold by three catalytic residues, which reserves the catalytic activity of HpaB. The distance from substrate to FAD is 4.1Å. The amine group of tyramine was hydrogen bound with side chain of Y212 and main chain of S462 (Figure 4C). The binding energy of tyrosol to mutant D11 is a little lower than that of tyramine, and the hydroxyl group at the tail of tyrosol has ability to hydrogen bond with Y212 and S462. However, the hydroxyl group in the side chain of Y212 could be negatively charged in the physical condition, it will prefer the amine tail of tyramine which is likely to be positively charged.

Introduction of tyrosine residues could be critical for recognition of tyramine, we also observed the tyrosine substitution at position 212 in mutant B7 (S210T, A211P and Q212Y) which preferred tyramine as a substrate (Figure 3F).

Mutant H7 displayed a dramatically higher catalytic activity on both tyrosol and tyramine, and the docking energy is also found to decrease greatly compared with those for wild-type HpaB, suggesting a significant increase in affinity of H7 for both tyrosol and tyramine. Compared with A10, introduction of M211 and G212 in H7 is likely to increase flexibility of the loop, which will allow entrance of either tyrosol or tyramine. In the docking models, both tyrosol and tyramine are hydrogen bounded with oxygen atom from FAD (Figure 4D, 4E). Moreover, hydroxyl of tyrosol can be also coordinated by side chain of T210, and main chains of G209 and M211 (Figure 4D). Notably, the distance from substrate to FAD decreased to 4.1Å, which may also enhance the catalytic efficiency of H7 for tyrosol and tyramine.

We have supplemented the results in page 10, paragraph 3 and Figure 4 in the revised manuscript.

4. Fig. 3D

"Fresh medium with substrate" and "Supernatant" should be written under the arrows.

Thank you, we have revised Fig. 3D according to your suggestion.

Reviewer #3 (Remarks to the Author):

In the manuscript from Chen et al, the authors engineered a promiscuous hydroxylase from wild type HpaBC acting on both tyrosol and tyramine substrates using a “directed divergent evolution” strategy. For that purpose, the authors developed a screening technique based on colorimetric differences. The resulting mutants were used to activate simultaneously two hydroxytyrosol biosynthetic pathways, lowering in that way the metabolic burden by flux redistribution.

Using saturation mutagenesis based on the HpaBC crystal structure from Thermus Thermophilus HB8, mutants were developed showing significantly higher activity either on tyrosol or tyramine, but not on both. Similarly, mutants that were active on both tyrosol and tyramine were screened.

The novelty is the use of directed evolution as a way of developing a mutant with promiscuous activity towards two substrates that can be simultaneously active in two alternative biosynthetic pathways.

The work presents convincing results. However, it is not completely clear how fluxes were distributed for mutant H7 (the highest producer with activity for both tyrosol and tyramine). Results shown in Figure 5C (time course) indicate no accumulation of intermediates, which was not previously observed in other mutants. Is there a way of investigating further how fluxes were distributed in the pathways?

Thank you for your suggestion.

To analyze the flux distribution, we determined the accumulations of intermediates in the very early stage (1-6 h) before hydroxytyrosol was formed. Tyramine (15 mM)

was used as a substrate for its good solubility compared with tyrosine. Samples were taken every hour from 1-6 h of culturing, and the concentrations of tyramine, dopamine, 3,4-DHPAA, 4HPAA, tyrosol and hydroxytyrosol were quantified. Tyrosol was the first intermediate to be detected from 1 h after inoculation. 4HPAA, the other intermediate in pathway 1, can be detected from 3 h. Nevertheless, dopamine and 3,4-DHPAA, the two intermediates in pathway 2, could not be detected until 6 h. At 6 h, dopamine accumulation was observed, and at the same time hydroxytyrosol showed up. However, 3,4-DHPAA was not detected all the time. The accumulations of 4HPAA and tyrosol disappeared from 6 h when hydroxytyrosol was observed (Table 1). It is possible that the H7 mutant became active enough after 5 h, simultaneously converting tyramine and tyrosol into dopamine and hydroxytyrosol, respectively, resulting in a rapid rise of hydroxytyrosol titer. We have supplemented the results in page 8, paragraph 5 and Table 1 in the revised manuscript.

REVIEWERS' COMMENTS:

Reviewer #1 (Remarks to the Author):

Although this research improved the activity of HpaBC towards different substrates via directed evolution, as mentioned before, biosynthesis of hydroxytyrosol has been reported at least by two different groups (Li et al. ACS Synthetic Biology, 2018, 7:647; Chung et al. Scientific Report. 2017; 7: 2578.). The pathway and enzymes used in this research are almost the same with Chung's work. About the designing multiple-pathway networks by using promiscuous enzymes, there are a lot of works already published during these ten years. For example, Lin et al. created two pathways to achieve production of caffeic acid by employing two promiscuous enzymes (Lin et al. Microbial Cell Factories. 2012, 11:42). For those consideration, this manuscript seems to need more novelty works to guarantee the publication in this journal.

Reviewer #2 (Remarks to the Author):

I consider that the authors corrected well the manuscript.
Toshiki Furuya

Reviewer #3 (Remarks to the Author):

In the revised manuscript, the authors have described the details of their directed divergent evolution technique to engineer a promiscuous hydroxylase from wild type HpaBC. The method has been successfully applied for tyrosol and tyramine. As raised in the comments from the reviewers, the authors acknowledge in the revision that there is low novelty in this application as others authors have addressed it before. However, the authors added in the revision a comparison with previously reported applications, highlighting the difference. The description of the flux calculations, which was another request from this reviewer, has been also included in the revised version.

In summary, the revised manuscript puts better into context their proposed method of directed divergent evolution which seems a potentially useful method for engineering promiscuous enzymes in future applications.

Many thanks for the comments. All pages and paragraphs refer to the revised manuscript. In the revised manuscript, the changes were indicated in yellow.

REVIEWERS' COMMENTS:

Reviewer #1 (Remarks to the Author):

Although this research improved the activity of HpaBC towards different substrates via directed evolution, as mentioned before, biosynthesis of hydroxytyrosol has been reported at least by two different groups (Li et al. ACS Synthetic Biology, 2018, 7:647; Chung et al. Scientific Report. 2017; 7: 2578.). The pathway and enzymes used in this research are almost the same with Chung' s work. About the designing multiple-pathway networks by using promiscuous enzymes, there are a lot of works already published during these ten years. For example, Lin et al. created two pathways to achieve production of caffeic acid by employing two promiscuous enzymes (Lin et al. Microbial Cell Factories. 2012, 11:42). For those consideration, this manuscript seems to need more novelty works to guarantee the publication in this journal.

Although several work on hydroxytyrosol biosynthesis using HpaBC enzyme were reported, no effort has been made in improving its low activity on tyrosol, including Chung's work. The enzymes used in this study are similar with those used in Chung's work, but we have developed the promiscuous tyrosol/tyramine hydroxylase activity in HpaBC and expanded this four enzyme-based pathway into two hydroxytyrosol biosynthetic routes. Thus although the enzymes are similar, the biosynthetic routes are not exactly same. In addition, pathway 2 was not mentioned in Chung's work.

We have mentioned that in nature some steps of natural product biosynthetic pathways occur in alternative orders, based on the substrate promiscuity of pathway enzymes (page 3, paragraph 2). Natural promiscuous enzymatic activities inevitably lead to multiple ways of biosynthesis, such as the caffeic acid biosynthesis in Lin's work. However, instead of using natural enzymes, in this work we intentionally designed the multiple pathways which does not exist in nature, taking advantage of the HpaBC mutants obtained via directed evolution. We intended to put forward the strategy of designing multiple pathways based on artificial promiscuous enzymes in laboratory engineering to improve biosynthetic efficiency, which is potentially applicable in natural product biosynthetic pathway engineering, and directed divergent evolution is a practical way to develop the promiscuous enzymes. So in this study we mimicked the natural strategy in metabolic engineering and significantly improved hydroxytyrosol biosynthetic efficiency.

Reviewer #2 (Remarks to the Author):

I consider that the authors corrected well the manuscript.

Toshiki Furuya

Thank you for your comments.

Reviewer #3 (Remarks to the Author):

In the revised manuscript, the authors have described the details of their directed divergent evolution technique to engineer a promiscuous hydroxylase from wild type HpaBC. The

method has been successfully applied for tyrosol and tyramine. As raised in the comments from the reviewers, the authors acknowledge in the revision that there is low novelty in this application as others authors have addressed it before. However, the authors added in the revision a comparison with previously reported applications, highlighting the difference. The description of the flux calculations, which was another request from this reviewer, has been also included in the revised version.

In summary, the revised manuscript puts better into context their proposed method of directed divergent evolution which seems a potentially useful method for engineering promiscuous enzymes in future applications.

Thank you for your comments.